# Development of Smallpox Antibody Testing and Surveillance Following Smallpox Vaccination in the Republic of Korea

**DOI:** 10.3390/vaccines12101105

**Published:** 2024-09-26

**Authors:** Hwachul Shin, SangEun Lee, Myung-Min Choi, Hwajung Yi, Yoon-Seok Chung

**Affiliations:** Division of High-Risk Pathogens, Bureau of Infectious Disease Diagnosis Control, Korea Disease Control and Prevention Agency, Cheongju 28159, Republic of Korea; hcshin2507@korea.kr (H.S.); eunlse@korea.kr (S.L.); cmm2463@korea.kr (M.-M.C.); pobee@korea.kr (H.Y.)

**Keywords:** smallpox, *Vaccinia* virus, smallpox vaccine, antibody titer, ELISA, bioterrorism

## Abstract

**Background**: Despite its global eradication in 1977, smallpox remains a concern owing to its potential as a biological agent, thereby prompting the ongoing development and utilization of its vaccine. Vaccination with the *Vaccinia* virus induces immunity against variola virus, the causative agent of smallpox; however, this immunity does not extend to viruses of different genera within the *Poxviridae* family. In this study, we aimed to assess the efficacy of an enzyme-linked immunosorbent assay (ELISA) method utilizing *Vaccinia* virus and recombinant A27L antigen for detecting antibodies against smallpox. **Methods.** An analysis of the serum from 20 individuals pre- and post-vaccination with the CJ strain (CJ50300) revealed neutralizing antibodies, which were confirmed using the plaque reduction neutralization test (PRNT). The ELISA method, validated with a PRNT_50_ cut-off value of >4, exhibited a sensitivity and specificity of >95% and was particularly reactive with the inactivated virus. Furthermore, adherence to the smallpox vaccination policy revealed significant differences in *Orthopoxvirus* antibody levels among 300 individuals of different age groups. These findings highlight the reliability and efficacy of the ELISA method in detecting post-vaccination antibodies and contribute significantly to diagnostic methods to prepare for potential smallpox resurgence and bioterrorism threats.

## 1. Introduction

The variola virus, a highly contagious and lethal pathogen that causes smallpox, poses a significant global health threat. The development of the cowpox-based vaccination by Edward Jenner marked the beginning of the eradication of smallpox [1]. The World Health Organization (WHO) spearheaded a global eradication campaign in 1966, which was successfully concluded with the last naturally occurring case recorded in Somalia on 26 October 1977. Subsequently, on 26 October 1979, the WHO officially declared the eradication of smallpox, representing the first instance of a disease being eliminated through human efforts [2].

After this eradication, there was a consensus to destroy all variola virus samples preserved by various countries. However, owing to bioterrorism concerns, the virus samples were kept under WHO supervision. The virus has been preserved to develop vaccines and treatments, advance research on viral structures, prepare for potential bioterrorist threats, and maintain historical and scientific records. The variola virus is currently stored only at the Center for Disease Control and Prevention (CDC) in the United States and the State Research Center of Virology and Biotechnology in Russia [3].

The potential threat of bioterrorism using the smallpox virus is a genuine concern. The variola virus is easily cultivable, can be freeze-dried, and remains stable when protected from heat and ultraviolet light [4]. Notable past bioterrorism incidents include the 1984 Rajneeshee cult’s *Salmonella* attack in Oregon, which affected over 750 people, and the 2001 anthrax letters in the USA, causing 22 infections and 5 deaths. These events underscore the need for vigilance and preparedness. During the Cold War, a former official overseeing the Soviet Union’s biological weapons project reportedly cultivated a large amount of the variola virus in laboratories and conducted experiments for its potential deployment through missiles [5]. Consequently, the CDC has officially classified the variola virus as a Category A bioterrorism agent [6].

Specific antivirals, such as cidofovir [7] and tecovirimat (ST-246) [8], are under development and exhibit treatment efficacy; however, no definitive cure exists. Therefore, vaccination remains the sole preventive measure. Rapid diagnosis is crucial in containing the spread of the disease during a smallpox outbreak. Direct diagnostic methods, including polymerase chain reaction (PCR), are highly sensitive [9,10]; however, their reliance on the presence of the virus during infection limits their ability to confirm past infections in recovered individuals [11]. Conversely, serological testing for immune antibodies is effective for epidemiological assessments and retrospective population surveys, as virus-specific immunoglobulin G antibodies remain in the body for an extended period [12,13,14].

The variola virus, a member of the *Poxviridae* family, is oval-shaped, approximately 350 nm × 270 nm in size, with single, linear, double-stranded DNA and a genome length of approximately 186 kb [15]. Poxviridae, which exclusively infects vertebrates, comprises eight genera. The variola virus belongs to the genus *Orthopoxvirus*, which also includes cowpox virus, *Vaccinia* virus, and monkeypox virus, which are capable of infecting humans. Notably, all poxviruses infecting vertebrates have common core antigens within the same genus, leading to their serological cross-reactivity [16]. Immunity developed through vaccination with the *Vaccinia* virus, which belongs to the same *Orthopoxvirus* genus as the variola virus, also confers an immune response to the variola virus. However, this immunity does not extend to viruses of different genera within the *Poxviridae* family [17]. Therefore, to address this gap in the literature, we sought to develop an enzyme-linked immunosorbent assay (ELISA) test using less lethal *Vaccinia* virus antibodies against the bioterrorism pathogen variola virus in humans and assess the effectiveness of this method in evaluating the efficacy of smallpox vaccines and antibody therapies, monitoring outbreaks, and determining population-level antibody titers. This approach aims to advance the capacity to effectively respond to potential bioterrorism threats involving smallpox.

## 2. Materials and Methods

### 2.1. Samples

The samples utilized in this study were acquired from 20 participants recruited through the “2021 Smallpox Vaccine Administration Training and First Response Personnel Vaccination Project”. Serum samples were collected at two time points—pre- and post-vaccination (4 weeks post-vaccination)—as negative and positive controls, respectively. The vaccination was administered using the CJ50300 strain vaccine, which was developed by Inno.N and is the vaccine likely used in Korea. Additionally, serum samples from 200 individuals born before 1978 (the year that smallpox vaccinations were stopped) and 100 individuals born thereafter were obtained from the Korea Disease Control and Prevention Agency (KDCA) National Biobank of Korea, collected through the “National Health and Nutrition Survey”. These samples were used to validate the performance of the developed ELISA test. The KDCA Institutional Review Board (IRB) approved this study (IRB number: 2021-06-08-3C-A), and all participants provided informed consent. Sample size calculation and power analysis were conducted to ensure the study was adequately powered to detect significant differences and validate the ELISA test performance.

### 2.2. Cells and Virus

Vero E6 (C1008) cells were procured from the American Type Culture Collection (ATCC, Manassas, VA, USA) and cultured at 37 °C in an atmosphere of 5% CO_2_ using Dulbecco’s modified Eagle’s medium (DMEM; Gibco, Grand Island, NY, USA) supplemented with 10% fetal bovine serum (FBS). The *Vaccinia* virus Western reserve (WR) strain (VR-1354™) was obtained from the ATCC and utilized for the study. Vero E6 cells at a density of 1.4 × 10^7^ cells were cultured in a T175 flask at 37 °C in 5% CO_2_ atmosphere using DMEM supplemented with 10% FBS. To facilitate handling and ensure the cells were in close contact with the flask surface, 3 mL of the medium was used, which is less than the typical volume for a T175 flask. After 1 day of culture, the medium was removed, and the stored *Vaccinia* virus (4 × 10^8^ plaque-forming units [PFU]/mL) was applied at a concentration of 0.5 multiplicity of infection using approximately 3 mL of DMEM with 1% FBS and allowed to adsorb for 2 days. Following infection, the supernatant was removed, and the cells were washed twice with phosphate-buffered saline (PBS) (Gibco) before adding 3 mL of DMEM with 1% FBS for three cycles of freeze–thawing. The dislodged cells were transferred to a 15 mL tube, and the virus-infected cells and culture medium were centrifuged at 700× *g* for 8 min at 37 °C to remove cell debris. The supernatant was then supplemented with 5 mL of DMEM and 1% FBS, aliquoted into 500 μL portions, and stored at −70 °C until further use.

### 2.3. Plaque Reduction Neutralization Test

The plaque reduction neutralization test (PRNT) was conducted by first diluting the inactivated serum (heated at 56 °C for 30 min) 1:2, followed by a series of 2-fold serial dilutions. Each dilution was then mixed in equal volumes of 100 μL with 200 PFU of Vaccinia virus, followed by neutralization at 37 °C for 1 h. The neutralized *Vaccinia* virus was then inoculated into Vero E6 cells cultured in a mono-layer in a prepared 12-well plate (Corning, Germany) and adsorbed for 1 h. After adsorption, the virus culture medium was removed. Furthermore, the overlay medium (2 × MEM with 10% FBS and 1% Penicillin-Streptomycin [250 mL], 2% Carboxymethyl cellulose [250 mL]) was added, followed by incubation at 37 °C in 5% CO_2_. When the plaque formation reached approximately 1 mm in diameter (approximately 54 h later), the overlay medium was removed, and the plaques were washed with 1× PBS at room temperature. The washed plates were then stained with a crystal violet mixture (1% Crystal Violet solution [200 mL], 37% formaldehyde [108 mL], 100% ethanol [25 mL], and distilled water [167 mL] to make up a total volume of 500 mL) for 30 min at room temperature, followed by another wash with 1× PBS. The neutralizing antibody titers (PRNT_50_) were determined as the reciprocal of the serum dilution that reduced virus plaques by ≥50% compared with that in the positive control (virus without neutralized serum). The experiments were conducted as singlets, with no replicates for each sample.

### 2.4. Anti-Vaccinia Virus IgG Antibody Detection ELISA

A carbonate–bicarbonate buffer (pH 9.4; Sigma, St. Louis, MA, USA) was dissolved at one capsule per 100 mL to prepare the coating buffer. The recombinant A27L antigen expressed in *Escherichia coli* (MBS1141613; MyBioSource, San Diego, CA, USA) was diluted to a final concentration of 0.2 μg/mL, and the inactivated WR *Vaccinia* virus was diluted to 3 × 10^6^ PFU/mL. The *Vaccinia* virus was inactivated by heating at 56 °C for 30 min prior to dilution. This method is commonly used to inactivate certain viruses. Both the inactivated virus and lysed cells were used as antigens in the ELISA. The diluted recombinant A27L antigen and inactivated WR Vaccinia virus were individually dispensed into Immulon^®^ 2HB 96-well microtiter ELISA plates (ImmunoChemistry Technologies, Bloomington, MN, USA) at 100 μL/well, sealed, and reacted for 24 h at 4 °C. The wells were washed five times with 350 μL/well of 1× PBS-Tween20 (PBST). Subsequently, 5% skim milk (232100; BD Difco, Franklin Lakes, NJ, USA) in 1× PBST was dispensed at 200 μL/well as a blocking buffer and reacted for 1 h at 37 °C. Post-blocking, the wells were rewashed before adding diluted human serum in the blocking buffer at 100 μL/well and incubating for 1 h at 37 °C. Thereafter, the wells were washed and incubated with anti-human immunoglobulin G (IgG) (H + L) and horseradish peroxidase conjugate (W4038; Promega, Madison, WI, USA) antibodies, diluted 1:10,000 in the blocking buffer, at 100 μL/well for 1 h at 37 °C. 

Subsequently, the wells were washed, and 3,3′,5,5′-tetramethylbenzidine (TMB) substrate solution (Thermo Fisher Scientific, Waltham, MA, USA) was added at 100 μL/well; the wells were then incubated at 37 °C for 10 min for color development. Subsequently, TMB Stop Buffer (GenDEPOT, Barker, TX, USA) was added at 50 μL/well to quench the reaction, and a plate reader (SpectraMax^®^ i3, Molecular Devices, San Jose, CA, USA) was used to record the values at a wavelength of 450 nm. The experiments were conducted as singlets, with no replicates for each wells.

### 2.5. Standardization of ELISA Results

Two methods were employed to standardize the variation in color development across plates. One method [18] (Method 1) involved establishing a blank well in each plate, computing the average value, and subsequently adding three times the standard deviation to derive the ΔO.D._1 value. The alternative method [19] (Method 2) involved subtracting the average O.D. value of normal human serum from the O.D. value of the sample to obtain ΔO.D._2. These approaches were used to normalize color development variations across plates.

### 2.6. Receiver Operating Characteristic Curve Analysis

The diagnostic accuracy of standardized ΔO.D. values was assessed using receiver operating characteristic (ROC) curve analysis with the MedCalc version 22 software (Belgium). The ROC curve analysis was used to determine the optimal cut-off value based on the relationship between sensitivity and specificity. The area under the ROC curve (AUC) represents the overall accuracy of the diagnostic test, with larger AUC values indicating enhanced accuracy. Ranging from 0 to 1, an AUC value closer to 1 signifies perfect diagnostic capability, whereas a value of 0.5 denotes performance equivalent to random guessing. This methodology facilitated the establishment of a cut-off value for the specific diagnostic test, evaluating its utility through the computed sensitivity and specificity at this threshold.

### 2.7. Statistical Analysis

Statistical analysis was performed using the GraphPad Prism version 9 software. Significant differences between the groups were determined using an independent sample *t*-test. This statistical test compares the means of two independent groups to assess if they are significantly different from each other. This analysis was based on the assumption that the data were normally distributed and focused on evaluating differences in mean values. In the *t*-test outcomes, statistical significance was set at a *p*-value of <0.05. All statistical analyses were performed using a two-tailed test, with data presented as mean ± standard deviation.

## 3. Results

### 3.1. Establishment of Standard Serum Using PRNT

The neutralizing antibody titers in the 20 smallpox vaccine recipients exhibited an increase, ranging between 4- and 64-fold in post-vaccination serum samples compared with that in pre-vaccination serum samples, indicating the development of neutralizing antibodies against the *Vaccinia* virus WR strain. The ROC curve analysis, using antibody titers from pre- and post-vaccination sera in the neutralization experiment, demonstrated that setting the neutralizing antibody titer against the *Vaccinia* virus at >4 yielded 100% sensitivity and specificity (Appendix A). Notably, all 20 pre-vaccination sera tested negative, whereas all 20 post-vaccination sera tested positive (Appendix A). These findings confirmed that sera collected pre- and post-smallpox vaccination could serve as a gold standard for positive and negative results, forming the basis for ELISA.

### 3.2. Results of IgG ELISA Using A27L Recombinant Antigen, a Common Antigen of Orthopoxvirus

Utilizing the A27L recombinant protein, a major antigen of *Vaccinia* virus, as the coating antigen in ELISA and diluting pre- and post-vaccination sera from the smallpox vaccine study 50–800-fold in two-fold increments for indirect ELISA, we revealed increased OD_450_ values in the post-vaccination serum samples compared with those in the pre-vaccination serum samples. However, these values required adjustment owing to potential variations in OD_450_ across different experimental plates caused by factors such as the TMB solution reaction time, experimental temperature, and humidity. Consequently, the ΔO.D._1 and ΔO.D._2 adjusted values were higher in the post-vaccination serum samples than in the pre-vaccination serum samples.

Furthermore, during analysis using the serum dilution factor, the highest AUC value was observed at a 50-fold dilution using the ΔO.D._2 adjustment method. Considering a positive result for cut-off points > 0.4667, both sensitivity and specificity were 100% (Appendix A), indicating the formation of smallpox vaccination-induced IgG antibodies against the A27L antigen. Therefore, the developed A27L IgG ELISA technique can measure the extent of IgG antibody formation against *Vaccinia* virus following smallpox vaccination.

### 3.3. IgG Titer Determination Using Vaccinia Virus ELISA

*Vaccinia* virus-infected Vero-E6 cells were lysed and used as inactivated antigens for ELISA. Pre- and post-vaccination sera from the smallpox vaccine study were diluted 50–800-fold in two-fold increments for indirect ELISA. After adjusting OD_450_, the ΔO.D._1- and ΔO.D._2-adjusted values of the post-vaccination serum samples were higher than those of the pre-vaccination serum samples. Additionally, during analysis using the serum dilution factor, the highest AUC value was observed at a 50-fold dilution using the ΔO.D._2 adjustment method. Considering a positive result for cut-off points > 0.0535, the sensitivity and specificity were 100% and 95%, respectively (Appendix A), indicating the formation of smallpox vaccination-induced IgG antibodies against *Vaccinia* virus. Therefore, the developed *Vaccinia* virus IgG titer can reflect the extent of IgG antibody formation against *Vaccinia* virus following smallpox vaccination.

### 3.4. Investigation of Smallpox IgG Titer Using Age-Specific Serum Samples from South Korea

Serum samples from the general population, collected through the National Health and Nutrition Survey in South Korea, were categorized into groups based on birth years and smallpox vaccination status as follows: Group A comprised individuals born between 1950 and 1961, who were likely vaccinated due to the smallpox vaccination campaigns actively conducted in South Korea until 1978, during which smallpox vaccination was mandatory for most of the population; Group B comprised individuals born between 1962 and 1970, who had a reduced likelihood of vaccination as smallpox was nearing global eradication and vaccination efforts were gradually declining; and Group C comprised individuals born between 1986 and 1995, who were least likely to have been vaccinated following the worldwide eradication of smallpox and the cessation of routine smallpox vaccinations. In this study, 100 serum samples were analyzed from each group (totaling 300 samples) for the PRNT and ELISA tests.

The PRNT results, with a cut-off for PRNT_50_ values set at >4, helped to detect neutralizing antibodies in the serum of 63 and 38 individuals from Groups A and B, respectively. No serum with neutralizing ability was identified in Group C. Using the same serum samples for the ELISA analysis with A27L recombinant antigen, 33 and 43 individuals from Groups A and B, respectively, tested positive. In Group C, two individuals tested positive. When using the *Vaccinia* virus to detect IgG antibodies, 44, 35, and 12 individuals from Groups A, B, and C, respectively, tested positive. However, when testing for immunoglobulin M (IgM) antibodies using the same antigen, individuals from all groups tested negative (Table 1). In this context, the “Positive Rate” refers to the percentage of individuals who tested positive for the presence of neutralizing or specific IgG antibodies in each group, while the “Negative Rate” refers to the percentage of individuals who tested negative. A statistical analysis of the ELISA results indicated that the averages for Groups A and B in the IgG ELISA were above the cut-off and that they significantly differed from the result of Group C, which was below the cut-off (*p* < 0.0001). These results are consistent with those obtained from the PRNT (Figure 1). The cut-off value of >4 for PRNT_50_ was established based on an ROC curve analysis, which validated its sensitivity and specificity in detecting neutralizing antibodies against the *Vaccinia* virus.

## 4. Discussion

The persistent threat of smallpox as a potential bioterrorism agent or biological weapon has caused South Korea to take proactive measures, including the accumulation of smallpox vaccine stockpiles and the development of vaccination plans for first responders and medical personnel. Furthermore, ensuring the scientific validation of vaccine efficacy, including the verification of antibody formation and their capacity to neutralize the virus, is imperative.

Herein, we present major findings from the serological analysis of smallpox vaccine efficacy using the PRNT and an ELISA analysis. In the absence of commercially available standard antibodies against the *Vaccinia* virus, sera were strategically collected from 20 individuals before and after vaccination with the CJ strain (CJ50300) smallpox vaccine as part of the “2021 Smallpox Vaccine Administration Training and First Response Personnel Vaccination Project”. The PRNT helped to validate the presence of neutralizing antibodies in all post-vaccination serum samples by establishing pre-vaccination sera as negative controls and post-vaccination serum samples, collected 4 weeks after vaccination, as positive controls, thus validating their use as standard controls.

In this study, a cut-off value of >4 for PRNT_50_ was established as the criterion for successful antibody generation against *Vaccinia* virus. The optimal cut-off for the IgG ELISA method was determined through an ROC curve analysis, using the PRNT results as the gold standard. The ELISA method demonstrated sensitivity and specificity of over 95% with the optimal cut-off, indicating the capability of the developed ELISA diagnostic method to discern the presence or absence of smallpox vaccine-induced antibodies.

In the IgG ELISA, using inactivated cell culture fluid from *Vaccinia* virus-infected cells and the recombinant A27L antigen expressed in *E. coli* as a coating antigen indicated that inactivated *Vaccinia* virus samples typically had higher average OD_450_. These results suggest that using the *Vaccinia* virus as an antigen results in better reactivity, attributable to the presence of various surface proteins and the corresponding antibodies in the serum. This approach offers enhanced sensitivity; however, the presence of multiple proteins from the infected Vero E6 cells may decrease its specificity.

The results of the analysis of 300 serum samples from the general population in South Korea, collected through the “National Health and Nutrition Survey Project”, aligned with those from the PRNT, confirming the persistent neutralizing capacity of the vaccine-induced antibodies. The absence of IgM antibodies in individuals of all groups indicated the lack of recent viral exposure or vaccination. These findings have crucial implications for public health, particularly regarding bioterrorism threats involving variola or related *Orthopoxvirus* species, highlighting the need for continued vigilance and the potential reevaluation of vaccination strategies.

The increased presence of neutralizing antibodies in individuals of Groups A and B, who were likely vaccinated due to the smallpox vaccination campaigns actively conducted in South Korea until 1978, during which smallpox vaccination was mandatory for most of the population, with no detectable neutralization in those from Group C, who were least likely to have been vaccinated because smallpox was officially declared eradicated in 1980 and routine smallpox vaccinations were no longer administered, indicated a generational immunity gap, likely owing to the discontinuation of smallpox vaccination post-1978. The older individuals in Groups A and B likely received the vaccine, whereas the younger individuals in Group C, who were born post-vaccination cessation, did not have these antibodies. These observations highlight notable public health concerns, particularly regarding potential smallpox resurgence, as it indicates heightened vulnerability of the younger, unvaccinated population in the event of an outbreak.

A recent CDC study on residual immunity from smallpox vaccination in China revealed that individuals born before 1980 who were tested for the *Vaccinia* virus Tiantan strain exhibited lasting neutralizing antibodies, suggesting long-lasting immunity and potential cross-immunity to related viruses in these individuals. Similarly, our findings from testing 300 individuals in South Korea showed a comparable pattern (Figure 2), indicating sustained immunity in older individuals vaccinated against smallpox. These findings further highlight the significance of our findings, as they may help prepare against emerging *Orthopoxvirus* threats. Furthermore, these findings highlight the substantial contribution of the developed ELISA method in monitoring vaccine-induced immunity and responding to potential bioterrorism threats [20].

This study has some limitations. First, its focus on a specific population in South Korea may restrict the generalizability of its findings to other regions with different demographics or vaccination histories. Second, the relatively small number of patients, particularly in specific analyses involving 20 individuals, could affect the statistical robustness and broader applicability of the results. A longitudinal study may be required to obtain comprehensive insights into the persistence of antibodies over time and potential decline in their levels.

In conclusion, in this study, we developed and optimized an ELISA method that can effectively detect *Orthopoxvirus* antibody responses across various population groups. This method could be used to identify heightened antibody responses in specific population groups, such as men who have sex with men (MSM), who may be at higher risk for infections such as mpox due to the cross-reactivity of antigens and antibodies within the *Orthopoxvirus genus*, which includes the *Vaccinia* and *Monkeypox* viruses. Although this study primarily focuses on *Vaccinia virus*-induced antibody responses, the cross-reactivity of *Orthopoxvirus* antigens, including those from the Monkeypox virus, suggests that the developed ELISA method could potentially be applied to detect antibody responses in individuals at risk for mpox. This highlights the broader applicability of the assay in assessing immune responses to various *Orthopoxvirus* infections. This aligns with recent research trends, highlighting the utility of the ELISA method in assessing antibody responses to different virus strains [21]. Furthermore, these results emphasize the importance of continuous monitoring and evaluation of *Orthopoxvirus* species in public health and highlight the significance of developing tailored vaccination strategies for specific population groups, especially for the MSM community. Finally, our findings reinforce the need for adaptable and targeted approaches to disease prevention and control.

## Figures and Tables

**Figure 1 vaccines-12-01105-f001:**
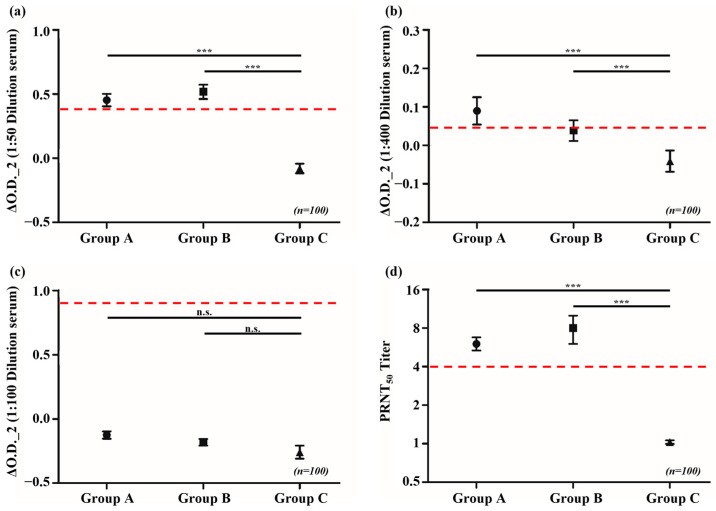
ELISA using serum from the National Health and Nutrition Survey. Upon examining the average ΔO.D._2 values of ELISA, a significant difference was observed in the IgG antibody levels between Groups A and B compared with the level in Group C. However, IgM antibodies were undetected in all groups. In all groups (A, B, and C), the sample size was *n* = 100. (**a**) Anti-A27L IgG ELISA, (**b**) anti-*Vaccinia* virus IgG ELISA, (**c**) anti-*Vaccinia* virus IgM ELISA, and (**d**) PRNT_50_ titer (the red dotted line represents the cut-off). PRNT, plaque reduction neutralization test; ELISA, enzyme-linked immunosorbent assay; IgG, immunoglobulin G; IgM, immunoglobulin M; **** p* < 0.001, “n.s.” indicates non-significant results, meaning there is no statistically significant difference between the compared groups.

**Figure 2 vaccines-12-01105-f002:**
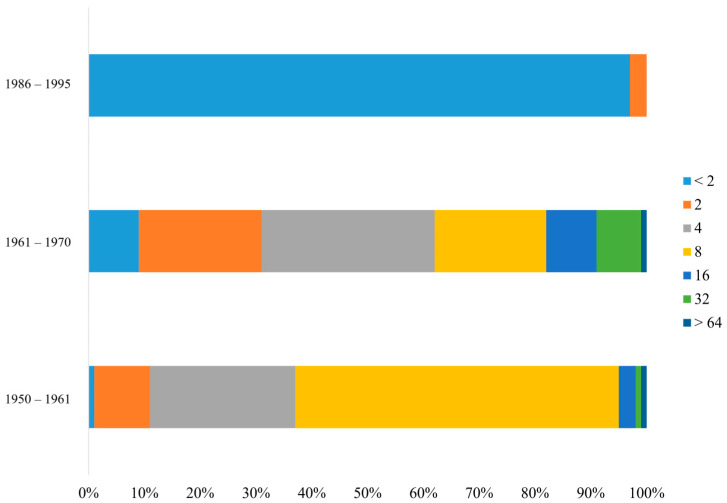
Distribution of neutralizing antibody titers against *Orthopoxvirus* by birth year. This bar graph illustrates the percentage of individuals within each birth year cohort and displays a range of neutralizing antibody titers measured using the PRNT. The titer values are categorized as <2, 4, 8, 16, 32, and >64, indicating the dilution factor at which the serum neutralized 50% of virus plaques, with a higher titer corresponding to enhanced neutralizing capability. Notably, most individuals in the 1950–1961 cohort exhibited high titers, whereas the 1986–1995 cohort showed a significant portion with titers < 2, suggesting a lack of neutralizing antibodies. PRNT: plaque reduction neutralization test.

**Table 1 vaccines-12-01105-t001:** Results of PRNT and ELISA analyses using serum from healthy individuals.

Group (Birth Year)	Analysis Technique	Positive Rate (%)	Negative Rate (%)	Total (%)
A (1950–1961)	PRNT	63	37	100
	A27L-IgG ELISA	33	67	100
	*Vaccinia*-IgG ELISA	44	54	100
	*Vaccinia*-IgM ELISA	0	100	100
B (1961–1970)	PRNT	38	62	100
	A27L-IgG ELISA	43	57	100
	*Vaccinia*-IgG ELISA	35	65	100
	*Vaccinia*-IgM ELISA	0	100	100
C (1986–1995)	PRNT	0	100	100
	A27L-IgG ELISA	2	98	100
	*Vaccinia*-IgG ELISA	12	88	100
	*Vaccinia*-IgM ELISA	0	100	100

PRNT, plaque reduction neutralization test; ELISA, enzyme-linked immunosorbent assay; IgG, immunoglobulin G; IgM, immunoglobulin M.

## Data Availability

The data are contained within the article or Appendix A.

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
