# Peer review of "Development of Smallpox Antibody Testing and Surveillance Following Smallpox Vaccination in the Republic of Korea"

_vaccines, 2024, doi:10.3390/vaccines12101105_

Round 1

Reviewer 1 Report

Comments and Suggestions for Authors

Overall, the paper is very well written and of interest to the scientific community.

Please review for minor grammatical errors throughout.

Under Section 2.1, please include information on CJ strain vaccine information.

On lines 112-113, it is not clear at what dilution and dilution scheme is used for the PRNT. It appears to be 1:2 followed by 2-fold dilutions, although this is not clear from the discussion.

On line 115, please include the monolayer confluency.

On lines 117-118, please confirm this is the correct formulation for overlay medium. This appears to be incorrectly stated, as plaques would not be able to form using this reagent as currently described.

On line 123, please indicate the reason for incubation for stain/inactivation at 37C.

On line 131, please define ‘They’.

For both section 2.3 and section 2.4, it is not clear how many replicates of each sample/dilution are tested. Were PRNT and ELISA samples analyzed in singlet, duplicate, etc?

The reviewer was not able to access Supplementary Tables and could not confirm information from lines 185, 186, 202.

Please clarify why minimal limits of detection for the assay are listed at two different PRNT50 values of 4 and 8 (lines 184, 229, 272).

Please confirm use of ‘smallpox vaccine sera’ (lines 192 and 208) as this is not clear if it is intended to reference the vaccine study pre- and post- samples, or possible vaccinated individuals from the larger sample study

Please add a statement on why the vaccine status of the individual cannot be confirmed (either by self-reporting, evidence of a scar, vaccine records, etc.), leading to use of the groupings of ‘likely vaccinated’, ‘reduced likelihood of vaccination’, and ‘least likely to have been vaccinated’. The use of these terms for assessing serum from individuals due to their vaccine status is not clear without confirmation of obtaining the vaccine (lines 222-226 and 295-297).

Please define Positive Rate and Negative Rate in Table 1 (line 245).

Please define which group is n=100 (or define n for remaining groups in plots) in Figure 1 (line 248).

PRNT titers are extremely low in Figure 1.d (line 153); however, it appears in another assay evaluating the second group of subjects the titers are much higher (Figure 2, line 319). Please clarify why the visualization of data is different and possibly prepare a similar figure for the data presented in Figure 1.

Following review of the ELISA method, there is no discussion of validation of the assay (controls, limits of detection, limits of quantification, linearity, etc). Please confirm use of the term validation is correct on line 328.

Please clarify how discussion of mpox and association of vaccination with VACV relates to use of the assay with monkeypox virus (lines 330-333). It is not clear how this statement contributes to the research and discussion of this study or allows for the assumption to be stated.

Comments on the Quality of English Language

Very few minor errors noted upon review. Overall, the manuscript is well-written and easy to follow. 

Author Response

Comments 1: Under Section 2.1, please include information on CJ strain vaccine information.
Response 1: Thank you for your valuable suggestion. We have now included the information regarding the CJ strain vaccine in Section 2.1. Specifically, we have added the following sentence in lines 84-86: "The vaccination was administered using the CJ50300 strain vaccine, which was developed by Inno.N and is the vaccine likely used in Korea." This addition provides clarity on the specific vaccine strain used in our study.
We appreciate your feedback in helping us improve the manuscript.

Comments 2: On lines 112-113, it is not clear at what dilution and dilution scheme is used for the PRNT. It appears to be 1:2 followed by 2-fold dilutions, although this is not clear from the discussion.
Response 2: To address the clarity issue, we have revised the description of the dilution scheme used for the PRNT in lines 114-117. The revised text now reads: "The plaque reduction neutralization test (PRNT) was conducted by first diluting the inactivated serum (heated at 56°C for 30 min) 1:2, followed by a series of 2-fold serial dilutions. Each dilution was then mixed in equal volumes of 100 μL with 200 PFU of Vaccinia virus, followed by neutralization at 37°C for 1 hour." We believe this clarification better explains the dilution process.

Comments 3: On line 115, please include the monolayer confluency.
Response 3: Thank you for pointing out the omission. In response to your comment, we have added information regarding the monolayer confluency in lines 117-119. The revised text now states: "The neutralized Vaccinia virus was then inoculated into Vero E6 cells cultured in a monolayer in a prepared 12-well plate (Corning, Germany) and adsorbed for 1 h." This addition provides a more detailed description of the cell culture conditions used in the assay.

Comments 4: On lines 117-118, please confirm this is the correct formulation for overlay medium. This appears to be incorrectly stated, as plaques would not be able to form using this reagent as currently described.
Response 4: Thank you for your observation. Upon review, we have identified that the formulation of the overlay medium in the original text was incorrect due to human error. We have corrected the description in lines 119-121. The revised text now reads: "After adsorption, the virus culture medium was removed. Furthermore, the overlay medium (2x MEM with 10% FBS and 1% Penicillin-Streptomycin [250mL], 2% Carboxymethyl cellulose [250mL]) was used." This correction ensures that the proper reagent formulation for plaque formation is clearly stated.
We revised the maniscript to clarify the points you have mentioned.

Comments 5: On line 123, please indicate the reason for incubation for stain/inactivation at 37C.
Response 5: There are no specific temperature conditions for incubation. The reaction was carried out at room temperature, so it seems there was a mistake.

Comments 6: On line 131, please define ‘They’.
Response 6: We have clarified the reference to "They" in line 139. The revised text now specifies: "The diluted recombinant A27L antigen and inactivated WR Vaccinia virus." This modification ensures that the subject is clearly defined for better readability and understanding.

Comments 7: For both section 2.3 and section 2.4, it is not clear how many replicates of each sample/dilution are tested. Were PRNT and ELISA samples analyzed in singlet, duplicate, etc?
Response 7: We have clarified the number of replicates for the PRNT and ELISA experiments in sections 2.3 and 2.4. The revised text in lines 130-131 now states: "The experiments were conducted as singlets, with no replicates for each sample." This addition provides clarity on the experimental design and analysis of the samples.

Comments 8: The reviewer was not able to access Supplementary Tables and could not confirm information from lines 185, 186, 202.
Response 8: We sincerely apologize for the omission of the supplementary file. It seems there was an error during the upload process.

Comments 9: Please clarify why minimal limits of detection for the assay are listed at two different PRNT50 values of 4 and 8 (lines 184, 229, 272). 
Response 9: Thank you for pointing this out. Upon review, we found that the correct PRNT50 value should be >4, and the discrepancy between the values of 4 and 8 was due to an error that occurred during translation and restructuring of the sentences. We have now corrected all instances to reflect the correct value of >4.
We revised the maniscript to clarify the points you have mentioned.

Comments 10: Please confirm use of ‘smallpox vaccine sera’ (lines 192 and 208) as this is not clear if it is intended to reference the vaccine study pre- and post- samples, or possible vaccinated individuals from the larger sample study
Response 10: Thank you for your observation. We have revised the wording to clarify the reference to "smallpox vaccine sera" in lines 200 and 217. The text now states: "Pre- and post-vaccination sera from the smallpox vaccine study," providing a clearer description of the samples being referenced.
We appreciate your attention to this detail and your helpful feedback.

Comments 11: Please add a statement on why the vaccine status of the individual cannot be confirmed (either by self-reporting, evidence of a scar, vaccine records, etc.), leading to use of the groupings of ‘likely vaccinated’, ‘reduced likelihood of vaccination’, and ‘least likely to have been vaccinated’. The use of these terms for assessing serum from individuals due to their vaccine status is not clear without confirmation of obtaining the vaccine (lines 222-226 and 295-297).
Response 11: Thank you for your thoughtful comment. We agree with your observation and have revised the text to provide clarification. Since the sera were obtained from the general population, only the birthdates of individuals were provided, and no additional personal information, such as vaccination records or physical evidence of vaccination (e.g., a scar), could be collected due to privacy protections. Therefore, we have revised the relevant section to read: "Groups A and B, who were likely vaccinated due to the smallpox vaccination campaigns actively conducted in South Korea until 1978, during which smallpox vaccination was mandatory for most of the population, with no detectable neutralization in those from Group C, who were least likely to have been vaccinated because smallpox was officially declared eradicated in 1980, and routine smallpox vaccinations were no longer administered."

Comments 12: Please define Positive Rate and Negative Rate in Table 1 (line 245).
Response 12: To clarify the terms "Positive Rate" and "Negative Rate" in Table 1, we have added the following explanation in lines 248-250: "In this context, the 'Positive Rate' refers to the percentage of individuals who tested positive for the presence of neutralizing or specific IgG antibodies in each group, while the 'Negative Rate' refers to the percentage of individuals who tested negative." We hope this addition provides better clarity for the reader.

Comments 13: Please define which group is n=100 (or define n for remaining groups in plots) in Figure 1 (line 248).
Response 13: Thank you for your comment. To clarify the sample size in Figure 1, we have added the following statement in line 266: "In all groups (A, B, and C), the sample size is n=100." This clarification ensures that the sample size for each group is clearly defined in the figure.

Comments 14: PRNT titers are extremely low in Figure 1.d (line 153); however, it appears in another assay evaluating the second group of subjects the titers are much higher (Figure 2, line 319). Please clarify why the visualization of data is different and possibly prepare a similar figure for the data presented in Figure 1.
Response 14: Figure 1.d presents the PRNT50 titer values as representative mean values for each group, while Figure 2 provides a more detailed distribution of titer values across a range of categories. The discrepancy in visualization arises due to the different approaches in representing the data: one focuses on average titer values, and the other on the full spectrum of titer distributions within each group.

Comments 15: Following review of the ELISA method, there is no discussion of validation of the assay (controls, limits of detection, limits of quantification, linearity, etc). Please confirm use of the term validation is correct on line 328.
Response 15: Thank you for your insightful comment. We fully agree with your observation, and upon review, we have determined that the term "Optimized" is more appropriate than "Validation." The text has been revised accordingly. This study focuses on the optimization of the ELISA assay and screening for immunogenicity in a larger population. We plan to conduct further studies on assay validation, including controls, limits of detection, quantification, and linearity, in the future.

Comments 16: Please clarify how discussion of mpox and association of vaccination with VACV relates to use of the assay with monkeypox virus (lines 330-333). It is not clear how this statement contributes to the research and discussion of this study or allows for the assumption to be stated.
Response 16: Thank you for your comment. We agree that the connection between mpox and Vaccinia virus vaccination could be more clearly articulated. To address this, we have revised the text in lines 351-356 as follows: "Although this study primarily focuses on Vaccinia virus-induced antibody responses, the cross-reactivity of Orthopoxvirus antigens, including those from Monkeypox virus, suggests that the developed ELISA method could potentially be applied to detect antibody responses in individuals at risk for mpox. This highlights the broader applicability of the assay in assessing immune responses to various Orthopoxvirus infections." This revision clarifies the relevance of the assay in the context of other Orthopoxvirus infections, including mpox.

We sincerely appreciate your thoughtful and thorough review of our manuscript. Your valuable insights and constructive feedback have been instrumental in enhancing the clarity and quality of our study. We are deeply grateful for your time and effort, and we look forward to further improving our work based on your suggestions. Thank you once again for your kind attention and consideration.

Reviewer 2 Report

Comments and Suggestions for Authors

The authors are conducting research to determine the possibility of using ELISA to assess the immune status of the population to smallpox. Vaccinia virus and recombinant bacterially expressed protein A27 were used as antigens to coat the plates. The virus neutralization test was used as a standard.

ELISA is a simple, reliable assay that requires less time and resources than the virus neutralization test.

The authors have put a lot of effort into preparing the manuscript, but there are a number of recommendations for its improvement.

Lines 101-106: The description of the method for producing the virus should be checked. The T175 flask requires a larger volume of medium than 3 ml.

Line 131: There is no clear description of the method for preparing the antigen based on the virus. How the virus was inactivated, was the inactivation carried out before or after its dilution. At the same time, further (line 207) the authors write that lysed cells infected with the virus were used as the inactivated antigen. Lines 134-138: It is recommended to use "wells" instead of "samples".

In the "Results" section, the authors refer to 4 supplementary  tables (lines 185, 186, 202, 214), but unfortunately, none of them were presented, which made it difficult to evaluate the results and conclusions obtained.

It is unclear how many samples were used in the tests. In "Materials and Methods", the authors talk about 300 serum samples. Comparing the information in the paragraph (lines 229-242) and Table 1, we can conclude that only 100 sera were analyzed.

Figure 1: Does the figure show data only from positive samples?

Author Response

Comments 1: Lines 101-106: The description of the method for producing the virus should be checked. The T175 flask requires a larger volume of medium than 3 ml.
Response 1: Thank you for your observation. We have revised the description in lines 102-103 to clarify the use of 3 mL of medium in the T175 flask. The revised text now states: "To facilitate handling and ensure the cells were in close contact with the flask surface, 3 mL of medium was used, which is less than the typical volume for a T175 flask." This adjustment reflects our intentional use of a smaller volume to improve cell contact and handling.

Comments 2: Line 131: There is no clear description of the method for preparing the antigen based on the virus. How the virus was inactivated, was the inactivation carried out before or after its dilution. At the same time, further (line 207) the authors write that lysed cells infected with the virus were used as the inactivated antigen. Lines 134-138: It is recommended to use "wells" instead of "samples".
Response 2: Thank you for your insightful comment. We apologize for not providing a more detailed description initially. In response to your feedback, we have revised the text in lines 136-139 to clarify the method for preparing the antigen: "The Vaccinia virus was inactivated by heating at 56°C for 30 minutes prior to dilution. This method is commonly used to inactivate certain viruses. Both the inactivated virus and lysed cells were used as antigens in the ELISA." We hope this revision offers better clarity regarding the antigen preparation process. And we have revised the text to replace 'samples' with 'wells' as per your suggestion.
We appreciate your careful review and apologize for the earlier lack of detail.

Comments 3: In the "Results" section, the authors refer to 4 supplementary tables (lines 185, 186, 202, 214), but unfortunately, none of them were presented, which made it difficult to evaluate the results and conclusions obtained.
Response 3: We sincerely apologize for the omission of the supplementary file. It seems there was an error during the upload process.

Comments 4: It is unclear how many samples were used in the tests. In "Materials and Methods", the authors talk about 300 serum samples. Comparing the information in the paragraph (lines 229-242) and Table 1, we can conclude that only 100 sera were analyzed.
Response 4: Thank you for your comment. To provide further clarity, we have added the following sentence in lines 238-239: "In this study, 100 serum samples were analyzed from each group (totaling 300 samples) for the PRNT and ELISA tests." This addition ensures that the number of samples analyzed is clearly communicated and consistent with the information presented in Table 1.

Comments 5: Figure 1: Does the figure show data only from positive samples?
Response 5: Thank you for your question. Figure 1 includes data from all 100 samples in each group, not only the positive samples. The figure presents the average values across all samples, including both positive and negative cases, to provide a comprehensive overview of the immune response within each group.

We would like to express our deepest gratitude for your thoughtful and meticulous review. Your insightful feedback has been invaluable in refining and enhancing the quality of our work. We truly appreciate the time and effort you have dedicated to improving this manuscript, and we look forward to continuing to learn from your suggestions. Thank you once again for your invaluable contribution.

Reviewer 3 Report

Comments and Suggestions for Authors

This submission aims to develop an ELISA for detecting antibodies against smallpox. The conclusion states that the assay can detect Orthopoxvirus antibodies, including those against Vaccinia and Monkeypox, but this is mentioned without supporting data.

It is unclear whether this study is focused solely on developing an ELISA for smallpox virus antibodies or if it also involves screening the population using this ELISA. If the study aims to address both objectives, the sample size is quite small for population screening and surveillance, and the validation and optimization of the ELISA are not well described.

There is no supplementary data submitted.

Intra-assay and inter-assay variability data are not provided.

ROC curve analysis data are not available.

No cross-reactivity studies have been conducted.

Author Response

Response: Thank you very much for your detailed feedback. This study serves as a preliminary investigation using 300 serum samples to develop and initially validate the ELISA method. While the sample size may be considered small, this research provides important insights and lays the groundwork for future, larger-scale studies. By conducting this initial screening with 300 samples, we were able to evaluate the feasibility of the method and predict the outcomes when applied to a larger population. We plan to expand the study with a larger sample size in future research to further validate and enhance the findings.

We also acknowledge the need for additional details regarding the validation and optimization of the ELISA method. ROC curve analysis data are available in the supplementary materials, and we will ensure that this data is uploaded. Intra-assay and Inter-assay variability studies, along with cross-reactivity studies, were not within the scope of this research but will be considered in future investigations.

Once again, thank you for your valuable comments. We look forward to providing the additional data to further support our findings.

Round 2

Reviewer 2 Report

Comments and Suggestions for Authors

The authors made many adjustments based on the recommendations of the reviewers.

The data in Table 1 and Figure 1 remain unclear to me. If we look at the data in the table, we see that according to the PRNT results, 37% of the samples in Group A were negative, i.e. the titer was less than 4 (cut off). At the same time, in Figure 1d, all the samples in Group A had a titer higher than 4. The situation is similar for the other analyses.

Author Response

주석 1: 표 1과 그림 1의 데이터는 여전히 불분명합니다. 표의 데이터를 살펴보면 PRNT 결과에 따르면 그룹 A의 샘플 중 37%가 음성이었습니다. 즉, 역가가 4 미만(차단)이었습니다. 동시에 그림 1d에서 그룹 A의 모든 샘플은 역가가 4보다 높았습니다. 다른 분석의 경우도 상황은 비슷합니다.

응답 1: 그림 1d에서 그래프는 100개 샘플의 평균 PRNT 역가와 해당 표준 편차(SD)를 나타내는데, 이는 각 샘플의 개별 양성 또는 음성 상태를 나타낸 표 1과 다를 수 있습니다. 표 1의 일부 샘플이 양성(PRNT50 > 4)으로 분류되더라도 정확한 PRNT50 값은 4~32 사이에서 달라질 수 있으며, 이는 그림 1d의 평균 값에 완전히 반영되지 않을 수 있습니다. 결과적으로 표 1과 그림 1d의 데이터 간에 차이가 발생할 수 있습니다. 데이터의 다양한 측면을 포착하기 위해 다차원 분석을 수행하여 다양한 관점에서 추세를 관찰했습니다.

귀중한 피드백과 사려 깊은 리뷰에 감사드립니다. 귀하의 의견을 신중하게 고려했으며 요청에 따라 설명을 제공했습니다. 귀하의 통찰력은 원고의 질을 개선하는 데 크게 기여했으며, 귀하의 시간과 노력에 큰 감사를 표합니다.

Reviewer 3 Report

Comments and Suggestions for Authors

Thanks for the reply. 

Author Response

[Comment 1] : Thanks for the reply. 

[Response 1] : Thank you very much for your time and thoughtful feedback. I have carefully addressed all the comments, and I believe the manuscript has significantly improved as a result. I am grateful for your guidance throughout this process, and I look forward to your final evaluation.